# Within- and across-frequency temporal processing and speech perception in cochlear implant users

**Chelsea M. Blankenship**[1]*, **Jareen Meinzen-Derr**[2], **Fawen Zhang**[3]

**1** Communication Sciences Research Center, Cincinnati Children's Hospital Medical Center, Cincinnati, Ohio, United States of America, **2** Division of Biostatistics & Epidemiology, Cincinnati Children's Hospital Medical Center, Cincinnati, Ohio, United States of America, **3** Department of Communication Sciences and Disorders, University of Cincinnati, Cincinnati, Ohio, United States of America

* chelsea.blankenship@cchmc.org

**Data Availability Statement:** All files are available from the Mendeley Data database (doi: 10.17632/ky3gcd6rpp.1).

**Funding:** This research was supported by the NIH NIDCD R15 DC011004 (Fawen Zhang) and the

# Abstract

## Objective

Cochlear implant (CI) recipient's speech perception performance is highly variable and is influenced by temporal processing abilities. Temporal processing is commonly assessed using a behavioral task that requires the participant to detect a silent gap with the pre- and post-gap stimuli of the same frequency (within-frequency gap detection) or of different frequencies (across-frequency gap detection). The purpose of the study was to evaluate behavioral and electrophysiological measures of within- and across-frequency temporal processing and their correlations with speech perception performance in CI users.

## Design

Participants included 11 post-lingually deafened adult CI users (n = 15 ears; Mean Age = 50.2 yrs) and 11 age- and gender-matched normal hearing (NH) individuals (n = 15 ears; Mean Age = 49.0 yrs). Speech perception was assessed with Consonant-Nucleus-Consonant Word Recognition (CNC), Arizona Biomedical Sentence Recognition (AzBio), and Bamford-Kowal-Bench Speech-in-Noise Test (BKB-SIN) tests. Within- and across-frequency behavioral gap detection thresholds (referred to as the GDT$_{within}$ and GDT$_{across}$) were measured using an adaptive, two-alternative, forced-choice procedure. Cortical auditory evoked potentials (CAEPs) were elicited using within- and across-frequency gap stimuli under four gap duration conditions (no gap, GDT, sub-threshold GDT, and supra-threshold GDT). Correlations among speech perception, GDTs, and CAEPs were examined.

## Results

CI users had poorer speech perception scores compared to NH listeners ($p < 0.05$), but the GDTs were not different between groups ($p > 0.05$). Compared to NH peers, CI users showed increased N1 latency in the CAEPs evoked by the across-frequency gap stimuli ($p < 0.05$). No group difference was observed for the CAEPs evoked by the within-frequency gap ($p > 0.05$). Three CI ears showing the longest GDT$_{within}$ also showed the poorest

University of Cincinnati Research Council (Chelsea Blankenship). The funders had no role in study design, data collection and analysis, decision to publish, or preparation of the manuscript.

**Competing interests:** The authors have declared that no competing interests exist.

**Abbreviations:** AzBio, Arizona Biomedical Sentence Test; BKB-SIN, Bamford-Kowal-Bench Speech-in-Noise Test; CAEP, Cortical Auditory Evoked Potential; CI, Cochlear Implant; CNC, Consonant-Nucleus-Consonant Word Test; EEG, Electroencephalography; GDT, Gap Detection Threshold; ICA, Independent Component Analysis; MCL, Most Comfortable Level; NH, Normal Hearing; SNR, Signal-to-Noise Ratio.

performance in speech in noise. The within-frequency CAEP increased in amplitude with the increase of gap duration; while the across-frequency CAEP displayed a similar amplitude for all gap durations. There was a significant correlation between speech scores and within-frequency CAEP measures for the supra-threshold GDT condition, with CI users with poorer speech performance having a smaller N1-P2 amplitude and longer N1 latency. No correlations were found among $GDT_{across}$, speech perception, and across-frequency CAEP measures.

## Conclusions

Within- and across-frequency gap detection may involve different neural mechanisms. The within-frequency gap detection task can help identify CI users with poor speech performance for rehabilitation. The within-frequency CAEP is a better predictor for speech perception performance than the across-frequency CAEP.

## Introduction

Temporal resolution, the ability to resolve rapid changes within the acoustic signal over time, is critical for speech understanding, music appreciation and sound localization [1–3]. A gap detection paradigm is a standard measure of auditory temporal resolution, which measures the shortest gap of silence between two acoustic markers that an individual can detect [4–10]. Previous researchers have commonly examined gap detection with the pre- and post-gap markers that are spectrally identical or similar, which is referred to as within-frequency gap detection [4, 11, 12]. Alternatively, gap detection can be assessed using the pre- and post-gap acoustic markers that are spectrally distinct, which is referred to as across-frequency gap detection [13, 14].

Gap detection thresholds (GDTs) are poorer in hearing-impaired listeners relative to normal hearing (NH) listeners due to the damage in the auditory nervous system [10]. In hearing-impaired listeners using a cochlear implant (CI), a prosthetic device converts sounds into electrical pulses that directly stimulate the auditory nerve. GDTs can be evaluated using different presentation modalities including acoustic stimulation and direct electrical stimulation (by passes the CI speech processor). Acoustic GDTs reflect the temporal processing impairments within the auditory system and signal distortion associated with the speech processing strategy and individual map parameters (stimulation rate, maxima, preprocessing strategies). Therefore, acoustic GDTs are expected to be poorer than those measured with direct electrical stimulation and are more representative of temporal processing abilities in everyday situations. Examining CI users' GDTs using acoustic stimulation may provide valuable information on why the clinical outcomes of cochlear implantation are highly variable across individuals.

CI users display a wide range of within-frequency gap detection threshold ($GDT_{within}$) with some individuals performing comparable to their NH peers (several ms) and others displaying much larger $GDT_{within}$ [15–19]. The correlation between $GDT_{within}$ and CI users' speech perception performance has demonstrated that larger $GDT_{within}$ is linked to poorer speech performance. Tyler et al. [20] reported the $GDT_{within}$ in CI users with a smaller $GDT_{within}$ ($< 40$ ms) displayed better speech perception performance than individuals with a larger $GDT_{within}$ ($> 40$ ms). Muchnik et al. [21] reported that CI users with open-set speech recognition had lower $GDT_{within}$ (Range:12–46 ms) than CI users without open-set speech recognition (GDT

Range:41–72 ms). The authors of the present study reported that CI recipients with smaller $GDT_{within}$ had better speech perception scores for words in quiet and sentences in noise [18].

In our everyday environment, sounds before and after silent gaps are rarely identical in frequency, therefore across-frequency gap detection thresholds ($GDT_{across}$) might be a measure that is more correlated to speech perception than the $GDT_{within}$. The $GDT_{across}$ has been reported to be higher than the $GDT_{within}$ [8, 13, 22–24], and it is highly variable even in NH listeners [23, 25]. The $GDT_{across}$ increases as the pre-gap marker duration is shortened [13, 17] and as the frequency separation is increased [7, 26, 27]. The $GDT_{across}$ in CI users has been examined in a limited number of studies, only using direct electrical stimulation [14, 28, 29]. For instance, van Wieringen and Wouters [28] reported that the $GDT_{across}$ was greater than 50 ms, which was much larger than the $GDT_{within}$ (< 5 ms). Hanekom and Shannon [14] reported that the $GDT_{across}$ increased when the channel separation between the markers increased and the $GDT_{across}$ (10–200 ms) was much larger than the $GDT_{within}$ (Range:1–4 ms).

Although behavioral GDTs are easily measured in adults, the tasks are not feasible for individuals who cannot perform behavioral tasks reliably, e.g., young children or adults with cognitive deficits. Thus, an objective electroencephalographic (EEG) measure that does not rely on behavioral cooperation but correlates well with behavioral GDTs would be beneficial to evaluate temporal processing in difficult-to-test populations [30]. Moreover, the combination of EEG and behavioral measures will explain possible neural mechanisms underlying gap detection and temporal processing, which is impossible if only behavioral measures alone are used, as in most previous studies [8, 13, 22, 28, 31].

The cortical auditory evoked potential (CAEP) recorded using EEG techniques has been examined using stimuli containing silent gaps in young adults with normal hearing [32–38], older adults with normal hearing or minimal hearing loss [36, 39, 40], individuals with auditory neuropathy [30, 32, 41] and cochlear implant recipients [41, 42]. When evoked by a stimulus containing a gap, the CAEP peaks, N1 and P2, occur at approximately 100 and 200 ms, respectively, after the onset of the pre- and post-gap markers (pre-gap CAEP and post-gap CAEP). In CI candidates with auditory neuropathy, He et al. [30] reported a significant negative correlation between open-set word recognition and the minimum gap in noise that can evoke a CAEP. In CI users, Mussoi and Brown [42] did not show a correlation between behavioral GDTs or CAEP evoked by electrical stimuli containing gaps and speech in noise performance.

Gap duration has a differential effect on the post-gap CAEP response amplitude and latency. With regard to the effect of gap duration on amplitude, it has been consistently reported that the post-gap CAEP amplitude increases as a function of the saliency of the acoustic change [32, 33, 35, 37, 38, 43]. In addition, gap durations that are inaudible or not behaviorally perceived (sub-threshold) generally do not elicit a post-gap CAEP response, and supra-threshold gap durations generate repeatable CAEP responses. Lastly, as the gap duration decreases towards threshold, the amplitude of the post-gap CAEP waveform subsequently decreases as well [37, 38]. With regard to the effect of gap duration on latency, the results are not as conclusive. Some studies report that gap duration had a significant effect on post-gap CAEP latency [33, 37], while other studies reported no effect [32, 38], or a non-monotonic effect on latency [35]. Pratt et al. [33] reported that the post-gap CAEP latency decreased as the gap duration increased. In contrast, Michalewski et al. [32] examined the effect of gap duration (2, 5, 10, 20 and 50 ms gaps) on post-gap CAEP latencies. They observed that while the 50 ms gap resulted in the longest latency compared to shorter gap duration conditions, results did not reach statistical significance. Palmer and Musiek [38] reported no change in post-gap CAEP latency as gap duration changed. Lastly, He et al. [35] reported that for gap durations up to 20 ms the post-gap CAEP latency tended to decrease, however, for longer gap durations the

post-gap CAEP latency increased. Taken together, results suggest the post-gap CAEP amplitude is a better indicator of auditory discrimination abilities and can be used as an objective indicator of the neural encoding of an acoustic change at the level of the auditory cortex.

In summary, the CAEP evoked by within-frequency gap stimuli is a promising objective tool to examine neural aspects of temporal processing. However, limited research on the CAEP has been conducted using across-frequency gap stimuli. Given the importance of temporal processing to speech perception, the present study was undertaken to: (1) examine the behavioral $GDT_{within}$ and $GDT_{across}$ and speech perception; 2) examine the CAEP evoked by within- and across-frequency gap stimuli; and (3) explore the relationship among behavioral GDTs, speech perception performance, and CAEP measures. The results would not only help understand the large variability in CI speech outcomes and but also result in the use of non-linguistic and objective measures to predict CI speech outcomes.

## Materials and methods

### Participants

Eleven adult CI users (Mean Age = 50.2 yrs; Age Range = 25.2–68.3) and eleven age- and gender-matched NH adults (Mean Age = 49.0 yrs; Age Range = 24.7–68.5) were enrolled in the study. Four CI users were bilaterally implanted and the rest were unilaterally implanted, with a total of 15 CI ears being tested individually. The NH listeners were tested in the same ear as their matched CI users, with a total of 15 NH ears being tested individually. All CI users were post-lingually deafened (onset of bilateral severe-to-profound hearing loss > 3 years of age), implanted with Cochlear Americas Implant System, (Cochlear Americas Ltd, New South Wales, Australia) and had a minimum of 1 year CI experience. CI users reported full time use of their device during all waking hours. Relevant demographic and device information is displayed in Table 1. All NH and CI participants were right-handed (Edinburgh Handedness Inventory; [44]), native speakers of American English, and did not report a history of neurological or psychiatric disorders, or brain injury. The research study was approved by the Institutional Review Board of the University of Cincinnati. Written informed consent was obtained from all participants and they were paid for participation.

### Stimuli

The stimuli used for behavioral gap detection tasks and EEG recordings were pure tones, consisting of a pre-gap marker, a silent gap, and a post-gap marker (Fig 1). The pure-tone markers (1 and 2 kHz) were created using Audacity software (version 1.2.5; open source, http://audacity.sourceforge.ent) with a sampling rate of 44.1 kHz. Pure-tone stimuli instead of more complex stimuli were used because: 1) stimulus complexity can affect GDTs [19, 45], and 2) the GDT assessed with pure tones is a more accurate measure of temporal resolution [4]. All stimuli were shaped with a $\cos^2$ window and included a 10 ms rise-fall time.

A silent gap was inserted between the pre- and post-gap markers and included a 1 ms rise-fall time around the silent gap, as in several previous studies [37, 43, 46]. Furthermore, the marker frequencies (1 and 2 kHz) were selected to facilitate comparison to studies that used a similar paradigm in adults with normal hearing or minimal hearing loss [37, 39]. The pre-gap marker was a 2 kHz tone for all stimuli. The post-gap marker was a 2 kHz tone for the within-frequency stimuli and a 1 kHz tone for the across-frequency stimuli. For the behavioral gap detection task, the pre and post-gap markers varied in duration from 250 to 350 ms to prevent the participant from using duration cues to aid in behavioral gap detection [7, 37]. For EEG testing, the pre and post-gap marker duration was fixed at 300 ms to allow averaging of the time-locked neural responses across stimulus presentations. An advantage of using a relatively

**Table 1. Cochlear implant recipient demographic and device information.**

| CI Participant | Sex | Ear | Etiology | AAT (yr) | AAO (yr) | LOIU (yr) | LOD (yr) | Internal Device | Processor | Strategy (Maxima) | Rate (pps/ch) | NH Age and Gender Match AAT (yr) |
|---|---|---|---|---|---|---|---|---|---|---|---|---|
| Sci10 | F | L | Hereditary | 61.6 | 27.0 | 2.5 | 32.0 | Nucleus CI24RE | Freedom | ACE (10) | 1800 | 59.6 |
| Sci18 | M | L | Congenital | 39.0 | 0.0 | 4.3 | 34.8 | Nucleus CI24RE | Freedom | ACE (10) | 900 | 36.0 |
| Sci19 | F | L | Fistulas | 25.2 | 4.0 | 4.0 | 17.3 | Nucleus 24 Contour Advance | Nucleus 5 | ACE (8) | 720 | 24.7 |
| Sci19 | F | R | Fistulas | 25.2 | 4.0 | 10.6 | 10.6 | N24M Straight | Nucleus 5 | ACE (10) | 900 | 24.7 |
| Sci36 | M | L | Unknown | 68.3 | 15.0 | 2.0 | 51.3 | Nucleus CI512 | Nucleus 5 | ACE (8) | 900 | 68.5 |
| Sci36 | M | R | Unknown | 68.3 | 15.0 | 2.2 | 51.2 | Nucleus CI512 | Nucleus 5 | ACE (8) | 900 | 68.5 |
| Sci39 | F | L | MMR | 45.5 | 4.0 | 2.6 | 39.0 | Nucleus CI512 | Nucleus 5 | ACE (8) | 900 | 44.3 |
| Sci39 | F | R | MMR | 45.4 | 4.0 | 2.0 | 39.5 | Nucleus 24 Contour Advance | Nucleus 5 | ACE (8) | 900 | 44.3 |
| Sci40 | F | R | Unknown | 54.0 | 35.0 | 6.3 | 12.7 | Nucleus CI24RE | Freedom | ACE (8) | 1200 | 57.6 |
| Sci41 | M | L | IV Antibiotics | 54.4 | 31.0 | 1.1 | 22.3 | Nucleus CI422 Slim Straight | Nucleus 5 | ACE (8) | 900 | 51.6 |
| Sci42 | M | L | Meniere's Disease | 68.0 | 25.0 | 2.7 | 40.2 | Nucleus CI512 | Nucleus 5 | ACE (8) | 900 | 63.6 |
| Sci43 | F | L | Otosclerosis/Noise Exposure | 59.7 | 35.0 | 1.5 | 23.2 | Nucleus CI422 Slim Straight | Nucleus 5 | ACE (8) | 900 | 59.3 |
| Sci44 | M | L | Gentamicin | 43.7 | 10.0 | 4.2 | 29.5 | Nucleus CI522 Slim Straight | Nucleus 6 | ACE (8) | 900 | 43.8 |
| Sci44 | M | R | Gentamicin | 43.7 | 10.0 | 2.7 | 31.0 | Hybrid L24* | Nucleus 6 | ACE (8) | 900 | 41.0 |
| Sci45 | F | L | Unknown | 50.3 | 5.0 | 7.8 | 37.5 | Nucleus CI512 | Nucleus 6 | ACE (8) | 900 | 50.3 |

*Note*. CI = Cochlear Implant; NH = Normal Hearing; MMR = Measles, Mumps, and Rubella; AAT = Age at Test; AAO = Age at Onset of Hearing Loss; LOIU = Length of Implant Use; LOD = Length of Auditory Deprivation; ACE = Advanced Combination Encoder

* Programmed as a traditional electrode array.

long duration for pre- and post-gap markers (300 ms) was that the CAEPs following the pre- and post-gap markers were less likely to overlap.

## General study procedures

All testing was completed in a double-walled sound treated booth (Industrial Acoustics Company, North Aurora, Illinois) over the course of one or two sessions depending on participant's preference and if one ear (3 hours of testing) or both ears were tested (6 hours of testing). All participants completed testing in the following order: audiological assessment, speech perception testing, behavioral GDT assessment, EEG testing. CI users completed all testing with their speech processors turned on and adjusted to their everyday settings (volume, sensitivity, and program) which were held constant throughout the entire test session(s). For NH individuals and unilateral CI users, testing was completed with the contralateral ear occluded with an E-A-R disposable foam ear plug. For bilateral CI users, testing was completed with the contralateral speech processor removed. All testing was completed at the Most Comfortable Level (MCL; 7 on a 0–10 loudness scale) [47]. Testing at the MCL has been commonly used in CI research and it allows easier comparison of test results between NH and CI users [4, 48, 49].

**Audiological assessment.** All NH listeners completed otoscopy and 226 Hz tympanometry to ensure normal outer and middle ear status. Next audiometric thresholds were measured (GSI-61 or AudioStar Pro; Grason Stradler, Eden Prairie, MN). For NH listeners, thresholds were measured at octave test frequencies from 0.25 to 8 kHz with pulsed pure-tones using

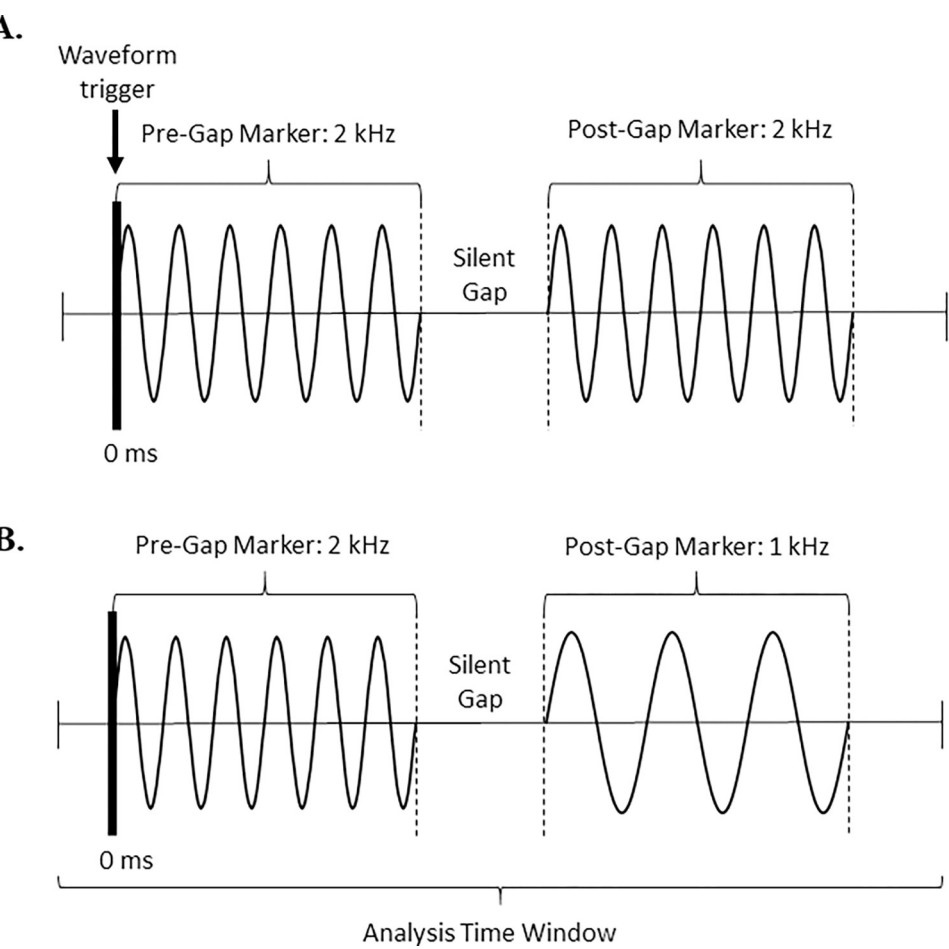

**Fig 1. Within- and across-frequency stimuli.** Schematic of within-frequency and across-frequency stimuli are shown in panel A and B respectively. Note, the figure is for visualization purposes only and does not reflect the rise-fall times outlined in the manuscript. For EEG testing, the recording was triggered at the pre-gap marker onset and the EEG analysis window included a 100 ms pre-stimulus period and extended 200 ms beyond on the offset of the post-gap marker.

insert ER-3A (Etymotic Research, Elk Grove Village, IL) earphones. For CI users, audiometric thresholds were measured using frequency-modulated tones from 0.25 to 6 kHz using sound field speakers (LSR305, Sweetwater, IN). The purpose of measuring hearing thresholds was to ensure the presentation level for the GDT assessment was at least 25–35 dB above threshold, which is necessary for optimal gap detection [6].

   **Speech perception assessment.** Speech perception performance was measured with the following commonly used clinical speech tests for adult CI Users [50–52]: 1) the Consonant-Nucleus-Consonant Word Test (CNC) assesses open-set monosyllabic word recognition in quiet. Two CNC word lists per test ear were administered to each tested ear; 2) the Arizona Biomedical Sentence Recognition Test (AzBio) assesses open-set sentence recognition in quiet and noise (AzBio-Quiet, AzBio-Noise). Participants were presented with two lists in quiet and two lists in noise (+10 dB signal-to-noise ratio; SNR) [52]; and 3) the Bamford-Kowal-Bench Speech-in-Noise Test (BKB-SIN) measures open-set sentence recognition with SNRs that ranged from +21 to -6 dB using an adaptive procedure. Two BKB-SIN word list pairs were administered [50]. All stimuli were presented at the MCL through either an insert earphone (NH

listeners) or a sound field speaker (LSR305, Sweetwater, IN) placed at 0 degrees azimuth 1 meter from the participant (CI users) with the contralateral ear occluded with an ear plug. Participants were instructed to verbally repeat what they heard and did not receive feedback based on their responses.

**Gap detection tasks.** Two separate gap detection tasks (within-frequency and across-frequency gap detection) were administered with a random order to assess the $GDT_{within}$ and $GDT_{across}$. Stimuli were presented using APEX software [53] on a laptop computer (Altec Lansing with SRS premium integrated soundcard) using an adaptive, two-alternative forced-choice procedure with an one-up one-down stepping rule. The gap durations ranged from 2 to 120 ms (2–20 ms in 2 ms increments, 20–120 in 5 ms increments). For each trial, the participant was presented with two sounds, one containing a silent gap (target) and the other no gap (reference). The no gap stimuli was a continuous 1 or 2 kHz tone that was constructed similarly to the target stimuli (e.g., $cos^2$ window and also included a 10 ms rise-fall time). The presentation order of the target and reference stimuli was randomized and the interval between presentations was 0.5 seconds. All stimuli were presented at the MCL through a loudspeaker placed at 0 degrees azimuth 1 meter from the participant with the contralateral ear occluded with an ear plug. For each trial, the participant was instructed to select the sound that contained a silent gap. All participants completed a practice trial first to ensure comprehension of the task. They were instructed to just focus on the gap and ignore the frequency change that might exist (in across-frequency gap detection). Visual feedback was provided, and testing continued until five reversals were completed and the mean of the last three reversals was recorded as the GDT.

**EEG recordings.** EEG recordings were conducted using a Neuroscan™ recording system (SCAN software version 4.3, Compumedics Neuroscan, Inc., Charlotte, NC) with a NuAmps digital amplifier. A 40-channel Neuroscan Quik-Cap (Compumedics Neuroscan, Inc., Charlotte, NC) was placed over the participant's scalp according to the 10–20 International system. The reference electrode was placed on the earlobe contralateral to the test ear, which has been found to reduce stimulus artifact in CI users [54, 55]. Continuous EEG activity was recorded with a sampling rate of 1000 Hz. Electrooculography was recorded to document eye movement activities, which were removed later during offline analysis. For CI users, approximately 1 to 3 electrodes surrounding the transmission coil were not used during recording, but the activities at these electrodes were interpolated during the offline analysis.

A total of eight EEG recordings were collected from each participant under four within- and four across-frequency conditions with gap durations that were customized based on the participants' $GDT_{within}$ and $GDT_{across}$. The four gap duration conditions included the following: (1) sub-threshold (GDT/3), (2) threshold (GDT), (3) supra-threshold (GDTx3), and (4) reference (no gap). Calculated gap durations were rounded to the nearest whole integer. The order of stimulus conditions was randomized across participants. Continuous EEG recordings were collected with a minimum of 200 and 400 stimulus trials collected from NH and CI users, respectively. The inter-stimulus interval was fixed at 0.9 seconds and triggers were time-locked to the onset of the pre-gap marker. For all participants, the EEG stimuli were presented at the MCL through a loudspeaker placed 1 meter from the test ear. During EEG testing, participants were seated in a comfortable chair, instructed to ignore the stimuli, relax, and read books of their choice.

## Data analysis

**Speech perception and gap detection data.** The CNC word (CNC-Word and CNC-Phoneme) and AzBio sentence (AzBio-Quiet, AzBio-Noise) tests were evaluated in terms of

percent correct. For the BKB-SIN, the SNR-50 was calculated to determine the SNR necessary to understand 50% of the key words contained in the sentence. The GDT$_{within}$ and GDT$_{across}$ were calculated as the mean gap duration (ms) of the last three reversals in the gap detection task.

**EEG data.** Initial EEG analysis was completed within Neuroscan software version 4.3. The EEG data was digitally band-pass filtered (0.1 to 30 Hz with a 6 dB/octave roll-off), segmented from 100 ms prior to the stimulus onset (-100 ms) to 200 ms beyond the offset of the post-gap marker, and baseline corrected using the pre-stimulus window (-100 to 0 ms). Subsequent analysis was completed in EEGLAB 13.6.5b (http://sccn.ucsd.edu/eeglab) running under Matlab R2017b (The Mathworks, Natick, MA), as in previous studies from our lab and other researchers [56–59]. Briefly, the data were visually checked to remove approximately 10% of the segments that contained non-stereotyped artifacts. Independent Component Analysis (ICA) was then used to decompose the EEG data into mutually independent components including those from neural and artifactual sources. Independent components that represented artifacts arising from ocular movement, electrode, or CI artifact were identified and removed after visual inspection of component properties including the waveform, 2-D voltage map, and the spectrum [60, 61]. The remaining components were then constructed to form the final EEG dataset. The unused channels were deleted and then interpolated. Data were re-referenced to the average reference [60, 62], baseline corrected using the pre-stimulus window (-100 to 0 ms), and filtered from 0.1 to 30 Hz using a band-pass Fast Fourier Transform linear filter.

Because the CAEP was most easily identified along the central/midline electrodes (Fz, FC3, FCz, FC4, Cz), data from these five electrodes were averaged together to form one final averaged waveform [32, 36]. For each tested ear, there were 8 averaged waveforms (4 gap conditions/stimulus type x 2 stimulus types). For each averaged waveform, the presence of pre- and post-gap CAEPs was determined by visual inspection of the waveform morphologies by two reviewers (Blankenship and Zhang). CAEP peak components (N1 and P2) were identified for both the pre- and post-gap markers in their specific latency ranges reported in previous studies [36, 63]. Specifically, the N1 was the maximum negative peak between 75 and 150 ms, and P2 was the maximum positive peak occurring between 150 and 220 ms after the onset of the markers. For convenient comparison, the post-gap CAEP peak latencies were adjusted so that the post-gap marker onset corresponded to time zero. The dependent variables obtained from EEG data were the N1-P2 peak-to-peak amplitudes (a commonly used measure for the CAEP amplitude [35, 38, 40] and N1 and P2 peak latencies.

**Statistical analysis.** Descriptive statistics were calculated for all outcome variables. Linear mixed effect models were used to examine group differences in behavioral and CAEP data. These models were selected because it is an appropriate method for analyzing non-independent data and allows the inclusion of both fixed and random effects. For behavioral measures, multiple linear mixed effect models were used to examine differences in speech perception and GDTs between the NH and CI users with participant group and test ear included as fixed effects. Age at test was included as a covariate and participant ID as a random effect to account for participants that were tested in both ears separately. For the CAEP data, multiple linear mixed effect models were conducted with participant group, gap duration condition, and test ear included as fixed effects. Age at test was included as a covariate and participant ID as a random effect. The Satterthwaite approximation method was used to estimate degrees of freedom and the Holm's step-down adjustment method was applied for pairwise comparisons. Bivariate scatter plots were used to evaluate the assumptions of linearity, multivariate normality, and homoscedasticity. Non-parametric spearman rank correlation coefficients were used to assess the strength of pairwise correlations between 1). GDT$_{across}$ and speech perception measures

(CNC, AzBio, BKB-SIN) and 2.) within- and across-frequency CAEP (N1-P2 amplitude, N1 and P2 latency) for the supra-threshold condition and speech perception (CNC-Word, AzBio-Noise, SNR-50). Bonferroni corrections were applied to account for multiple comparisons. Data were analyzed using SPSS statistical software (IBM Corp. Released 2019. IBM SPSS Statistics for Windows, Version 26.0. Armonk, NY: IBM Corp.).

## Results

### Audiometric thresholds

NH and CI group mean audiometric thresholds are displayed in Fig 2. NH individuals had thresholds $\leq$ 25 dB from 0.25 to 8 kHz. CI users had thresholds that ranged from 15 to 45 dB HL across all test frequencies. Obtaining thresholds ensured that the presentation level for GDT assessment was at least 25–35 dB above threshold [6].

### Speech perception and GDTs

For non-adaptive speech perception tests (CNC and AzBio), all NH listeners' scores were $\geq$ 97%; CI users showed variable performance (Range = 38–99%, with some CI users' showing much worse scores compared to NH listeners' scores while some were comparable to NH listeners. On the adaptive BKB-SIN test, the mean SNR-50 was -0.7 dB for NH listeners and much higher in CI users (+8.2 dB). On the gap detection tasks, all NH listeners had a GDT$_{within}$ of 2 ms, which was the smallest gap duration on the task. All CI users also had a GDT$_{within}$ of 2 ms, except for 3 CI ears (Sci36-Left = 51.7 ms, Sci36-Right = 26.7 ms, Sci43-Left = 41.7 ms). The GDT$_{across}$ had a similar range between NH (14.6–120.0 ms) and CI users

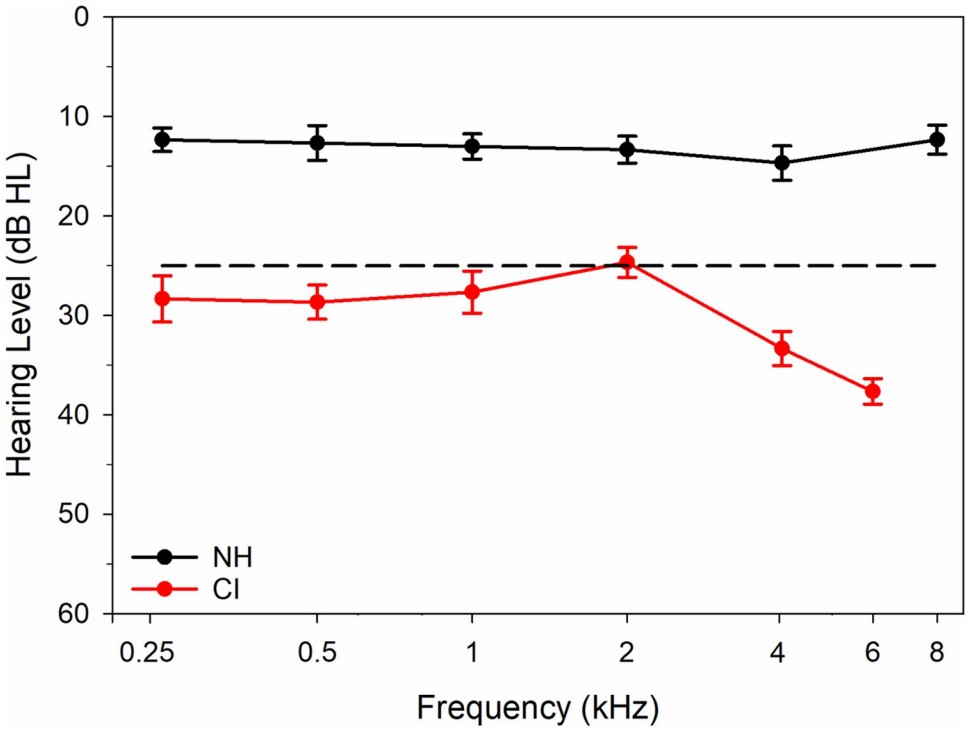

**Fig 2. NH and CI group mean audiometric thresholds.** Mean audiometric thresholds for NH (n = 15 ears) and CI recipients (n = 15 ears) at octave test frequencies from 0.25 to 8 kHz. The black dashed line indicates normal hearing criteria and error bars represent 95% confidence intervals.

(25.0–120.0 ms) but NH listeners showed a lower mean $GDT_{across}$. Group mean speech perception scores and GDTs are displayed in S1 Table.

Linear mixed effect model results (S2 Table) showed CI users performed significantly poorer ($p < 0.001$) on all individual speech perception measures compared to NH individuals but lacked a group difference in $GDT_{within}$ or $GDT_{across}$ ($p > 0.05$). Age at test was a significant factor that negatively affected $GDT_{across}$ ($p < 0.001$).

## EEG results

Group mean CAEP waveforms for within-frequency pre- and post-gap markers in NH listeners and CI users are displayed in Fig 3. The CAEP mean and standard deviation of the peak amplitude and latency are displayed in Table 2. It can be seen that CI users' CAEPs for both pre- and post-markers displayed smaller amplitudes and longer peak latencies compared to NH listeners. The post-gap marker CAEP has a smaller amplitude than the pre-gap marker CAEP for both groups. For NH listeners, there is a trend that the CAEP for the post-gap marker has a progressively decreased amplitude for shorter gap durations (Fig 3 top right panel). Such trend was not seen in CI users (Fig 3 bottom right panel).

Multiple mixed effect models were used to evaluate the effect of group, gap duration condition, test ear, and age at test on within-frequency post-gap CAEP amplitude and latency, which is the CAEP of interest in this study (S3 Table). For the N1-P2 amplitude, no group effect was observed. Instead, a significant effect of age at test indicated that older individuals (NH and CI) displayed smaller N1-P2 amplitudes (Mean = 1.2 μV) than younger adults (Mean = 0.5 μV, $p = 0.041$). There was no main effect of gap duration condition or test ear. For N1 and P2 latency, no effect of group, gap duration condition, test ear, or age at test was observed ($p > 0.05$).

Group mean across-frequency pre- and post-gap CAEP waveforms are displayed in Fig 4. The CAEP mean and standard deviation of N1-P2 amplitudes and peak latencies are displayed in Table 3. It can be seen that CI users' CAEPs for both pre- and post-markers displayed longer peak latencies compared to NH listeners, but the N1-P2 amplitude was similar for both groups, which did not differ for different gap duration conditions (Fig 4 top and bottom right panels). Moreover, the post-gap marker CAEP has a slightly larger amplitude than the pre-gap marker CAEP for both groups.

Multiple linear mixed effect models were used to evaluate the effect of group, gap duration condition, test ear, and age at test on the across-frequency post-gap CAEP amplitude and latency values (S3 Table). No significant effects of group, condition, test ear, or age at test were observed ($p > 0.05$) for N1-P2 amplitude. CI users displayed increased N1 latencies (Mean = 117.1 ms) compared to NH listeners (Mean = 107.8 ms, $p = 0.017$), and right ear latencies were increased (Mean = 116.7 ms) compared to the left ear (Mean = 108.2 ms, $p = 0.012$). Age at test was also significant for N1 latency ($p = 0.010$) and P2 latency ($p = 0.003$) with younger adults (NH and CI) displaying shorter latencies.

## GDTs vs. speech perception

The correlation between the $GDT_{within}$ and speech perception was not examined because all CI ears except 3 demonstrated a $GDT_{within}$ of 2 ms, the lowest gap duration in the within-frequency gap detection task. However, it was noted that these 3 CI ears with a much longer $GDT_{within}$ (Sci36-Left = 51.7 ms, Sci36-Right = 26.7 ms, Sci43-Left = 41.7 ms) have poorer speech performance, especially in noise (Median: AzBio-Noise = 51%, SNR-50 = +11 dB) compared to other CI ears (Median: AzBio-Noise = 76%, SNR-50 = +7 dB).

Non-parametric spearman rank correlation coefficients were used to assess the strength of pairwise correlations between $GDT_{across}$ and speech perception measures (CNC, AzBio,

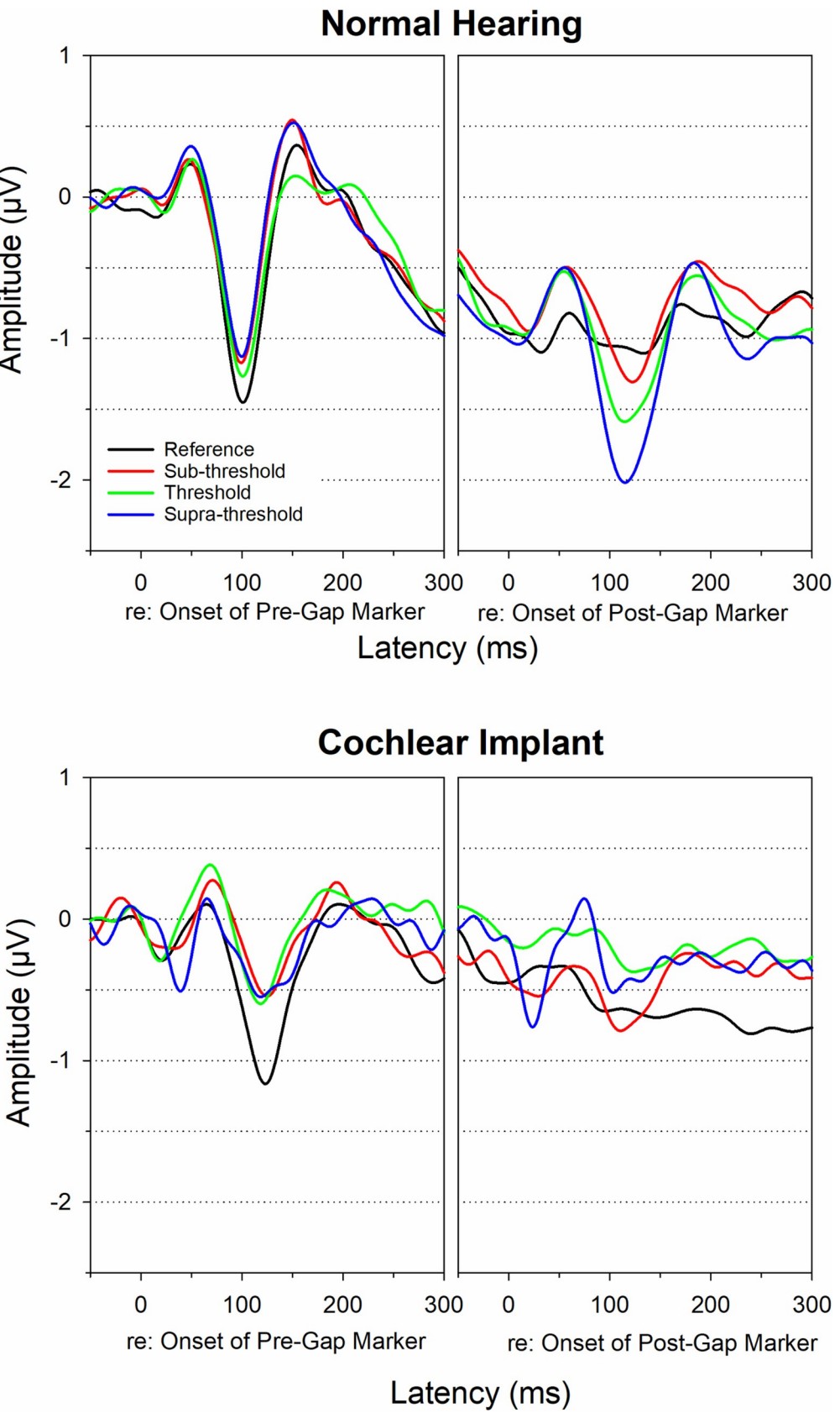

**Fig 3. NH and CI group mean within-frequency CAEP waveforms.** NH and CI group mean pre- and post-gap within-frequency CAEP waveforms for all gap duration conditions (reference, sub-threshold, threshold, supra-threshold).

BKB-SIN) for the NH and CI group collectively. Bonferroni corrections were applied to account for multiple comparisons, and therefore a *p*-value of 0.01 was considered statistically significant. None of the correlations reached significance following a Bonferroni correction ($p > 0.01$).

## CAEPs vs. speech perception

Non-parametric spearman rank correlation coefficients were used to assess the strength of pairwise correlations between within- and across-frequency supra-threshold post-gap CAEP (N1-P2 amplitude, N1 and P2 latency) and speech perception (CNC-Word, AzBio-Noise, SNR-50) for the NH and CI group collectively. Bonferroni corrections were applied to account for multiple comparisons, and therefore a *p*-value of 0.004 was considered statistically significant. Within-frequency correlations revealed a significant positive correlation between CNC-Word score and N1-P2 amplitude, a significant negative correlation between AzBio-Noise and N1 latency, and a significant positive relationship between the BKB-SIN SNR-50 and N1 latency. For across-frequency CAEPs, none of the correlations reached significance following a Bonferroni correction ($p>0.004$). The correlations between CAEP measures for the within-frequency supra-threshold gaps and speech perception are displayed in scatterplots in Fig 5.

## Discussion

The purpose of the present study was to examine the behavioral $GDT_{across}$ and $GDT_{within}$ and CAEPs evoked by within- and across-frequency gap stimuli; furthermore, the correlation between these measures and speech perception were explored. CI users had significantly

**Table 2. NH and CI group mean within-frequency CAEP amplitude and latency values.**

| Marker | Group | Condition | Count (n) | Amplitude (µV)* Latency (ms)* | | |
| --- | --- | --- | --- | --- | --- | --- |
| | | | | **N1-P2** | **N1** | **P2** |
| Pre-Gap | NH | Reference | 15 | 2.3 ± 1.4 | 100.7 ± 7.2 | 166.6 ± 24.7 |
| | | Sub-threshold | 15 | 2.0 ± 1.4 | 100.9 ± 8.9 | 160.5 ± 22.6 |
| | | Threshold | 15 | 2.0 ± 1.1 | 99.5 ± 7.5 | 154.9 ± 22.3 |
| | | Supra-threshold | 15 | 2.1 ± 1.2 | 99.9 ± 5.6 | 159.1 ± 23.9 |
| | CI | Reference | 15 | 1.8 ± 1.3 | 121.3 ± 15.3 | 188.5 ± 27.5 |
| | | Sub-threshold | 15 | 1.6 ± 0.9 | 116.7 ± 27.6 | 183.0 ± 24.6 |
| | | Threshold | 15 | 1.2 ± 0.7 | 123.3 ± 18.7 | 192.3 ± 43.9 |
| | | Supra-threshold | 15 | 1.6 ± 1.4 | 120.8 ± 21.2 | 187.4 ± 36.6 |
| Post-Gap | NH | Reference | — | — | — | — |
| | | Sub-threshold | 13 | 1.2 ± 0.7 | 126.1 ± 17.9 | 185.4 ± 14.4 |
| | | Threshold | 13 | 1.5 ±0.6 | 120.1 ± 12.8 | 192.7 ± 12.7 |
| | | Supra-threshold | 15 | 1.8 ± 1.1 | 120.7 ± 12.2 | 187.1 ± 15.4 |
| | CI | Reference | — | — | — | — |
| | | Sub-threshold | 13 | 1.2 ± 1.0 | 136.2 ± 41.4 | 190.7 ± 42.3 |
| | | Threshold | 13 | 1.0 ± 1.1 | 146.4 ± 43.1 | 207.8 ± 57.8 |
| | | Supra-threshold | 14 | 1.2 ± .9 | 157.5 + 46.8 | 206.8 ± 47.0 |

* Mean values ± one standard deviation

## Normal Hearing

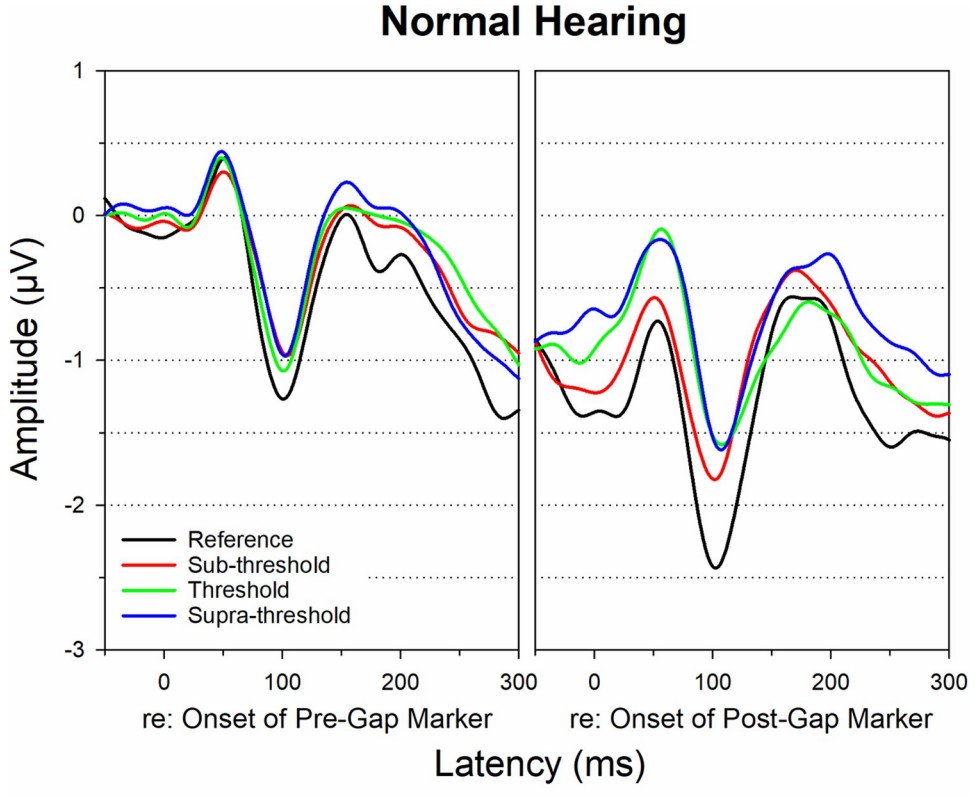

## Cochelar Implant

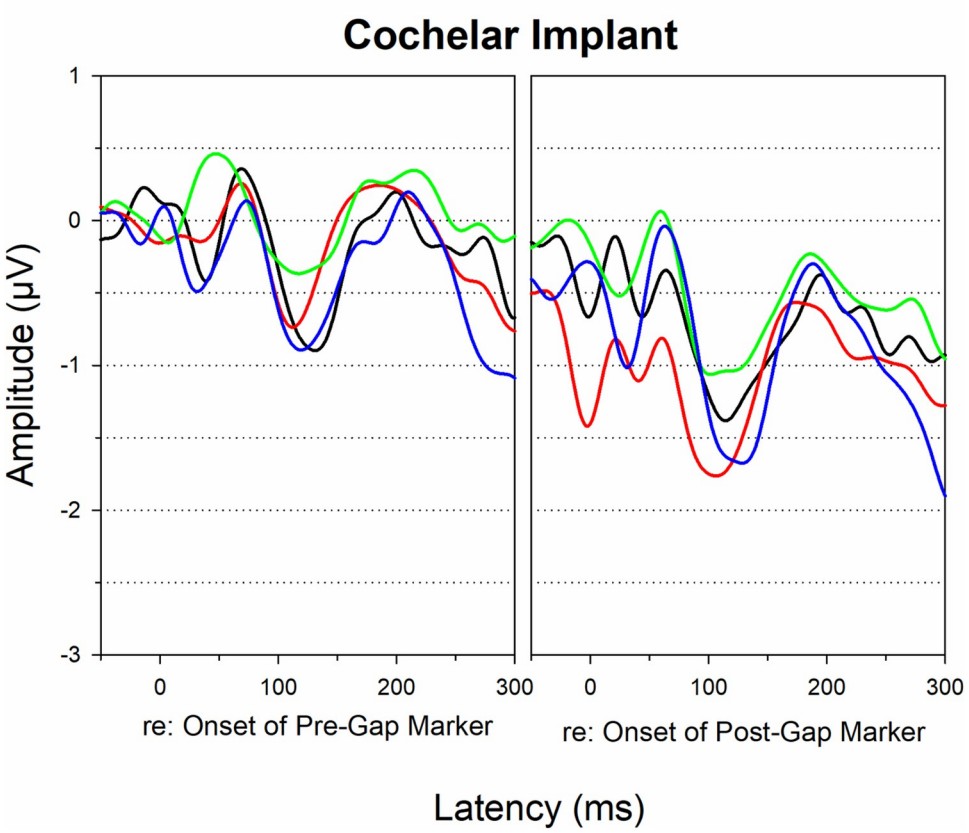

**Fig 4. NH and CI group mean across-frequency CAEP waveforms.** NH and CI group mean pre- and post-gap across-frequency CAEP waveforms for all gap duration conditions (reference, sub-threshold, threshold, supra-threshold).

poorer speech perception scores, but their $GDT_{across}$ and $GDT_{within}$ were not significantly different from NH listeners. Age at test negatively affected $GDT_{across}$ and within- and across-frequency CAEPs. Additionally, across-frequency CAEP results displayed a significant group effect for N1 latency and an ear effect for N1 latency.

## Speech perception

NH listeners performed near ceiling level in speech tasks while CI users performed significantly poorer on all speech perception tasks. On the CNC and AzBio, NH listeners displayed mean scores that were > 99%. CI users that displayed mean scores that ranged from 66 to 89% correct, which is on average 11 to 34% poorer compared to NH participants. On the BKB-SIN, CI users required an 8.9 dB SNR increase compared to NH participants to understand 50% of the target words in the sentence.

NH and CI group mean speech perception scores in this study are consistent with those reported in the literature [18, 50, 64]. In a previous study by the authors, a mean CNC-Word score of 64.9%, CNC-Phoneme score of 79.1%, and AzBio-Quiet of 75.5% was reported in a group of post-lingually deafened CI users [18]. Gifford et al. [65] reported mean speech recognition scores for post-lingually deafened adult CI users for the unilateral listening condition were: 55.7% for CNC-Word, 72.1% for AzBio-Quiet, and +11.4 dB for BKB-SIN SNR-50. Donaldson et al. [48] reported a median BKB-SIN SNR-50 value of +11.9 dB in a group of adult CI users. Slight differences observed in speech perception scores across studies could be due to several factors including differences in stimulus presentation levels, participant age, and the number of unilateral and bilateral CI recipients in the study.

**Table 3. NH and CI group mean across-frequency CAEP amplitude and latency values.**

| Marker | Group | Condition | Count (n) | Amplitude (µV)* | Latency (ms)* | |
| --- | --- | --- | --- | --- | --- | --- |
| | | | | N1-P2 | N1 | P2 |
| Pre-Gap | NH | Reference | 15 | 1.8 ± 1.1 | 101.6 ± 10.8 | 165.0 ± 27.0 |
| | | Sub-threshold | 15 | 1.5 ± 0.8 | 101.5 ± 14.1 | 165.0 ± 26.0 |
| | | Threshold | 15 | 1.9 ± 1.0 | 97.5 ± 10.1 | 167.4 ± 31.5 |
| | | Supra-threshold | 15 | 1.7 ± 0.8 | 101.1 ± 9.6 | 170.0 ± 25.7 |
| | CI | Reference | 15 | 1.7 ± 1.3 | 123.6 ± 16.6 | 179.4 ± 19.4 |
| | | Sub-threshold | 15 | 1.7 ± 0.9 | 116.4 ± 18.5 | 178.3 ± 31.2 |
| | | Threshold | 15 | 1.5 ± 1.3 | 116.5 ± 22.9 | 171.2 ± 36.6 |
| | | Supra-threshold | 15 | 1.8 ± 1.1 | 125.7 ± 17.4 | 176.7 ± 24.9 |
| Post-Gap | NH | Reference | 15 | 2.4 ± 1.1 | 103.1 ± 10.0 | 177.4 ± 18.4 |
| | | Sub-threshold | 15 | 2.0 ± 1.0 | 102.8 ± 10.4 | 172.8 ± 23.0 |
| | | Threshold | 14 | 1.4 ± 0.9 | 108.8 ± 18.3 | 178.9 ± 24.4 |
| | | Supra-threshold | 15 | 1.7 ± 0.7 | 109.2 ± 13.9 | 177.6 ± 24.0 |
| | CI | Reference | 15 | 1.9 ± 1.5 | 115.5 ± 23.8 | 175.2 ± 34.1 |
| | | Sub-threshold | 13 | 2.0 ± 1.1 | 111.3 ± 14.4 | 180.3 ± 18.9 |
| | | Threshold | 14 | 1.9 ± 1.5 | 118.1 ± 26.9 | 183.4 ± 43.1 |
| | | Supra-threshold | 15 | 2.2 ± 1.6 | 121.3 ± 18.6 | 184.4 ± 24.8 |

*Mean values ± one standard deviation.

## Within Frequency - Suprathreshold CAEP

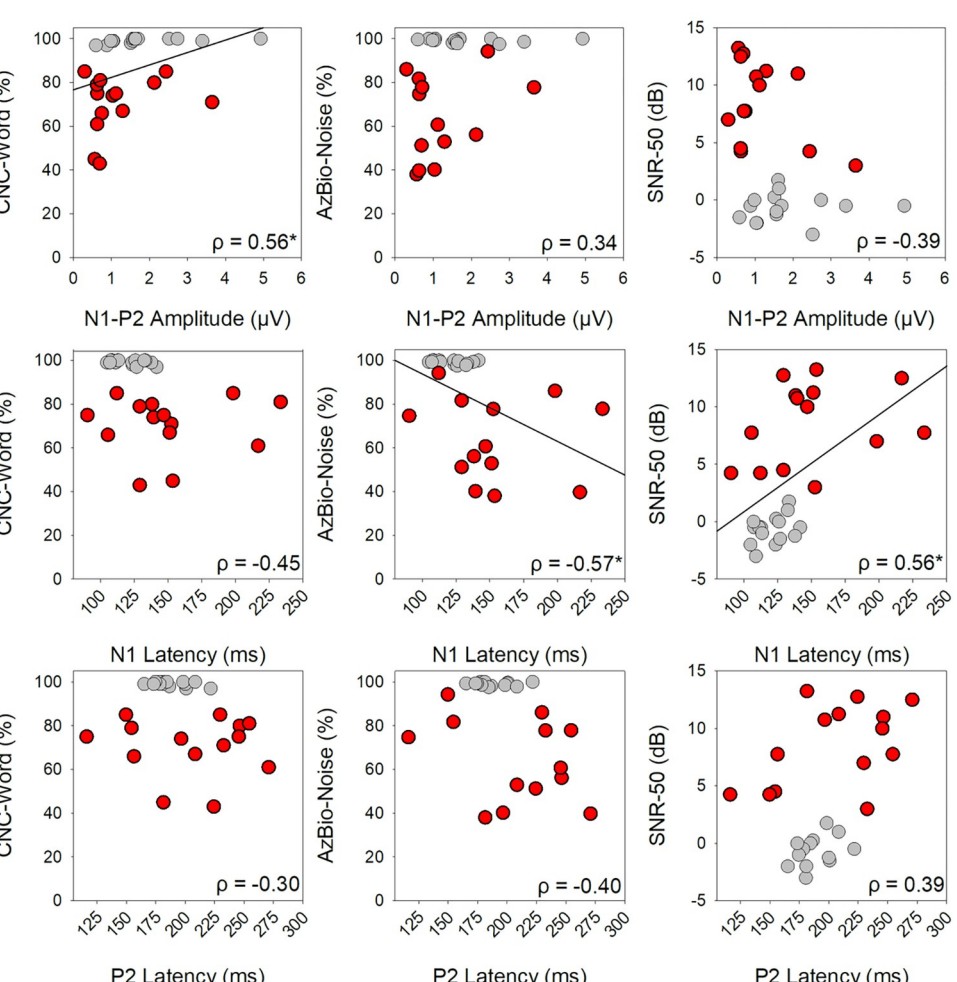

**Fig 5. Speech perception vs. CAEP amplitude and latency scatter plots.** Speech perception scores plotted as a function of within-frequency supra-threshold CAEP amplitude and latency. NH participants are displayed in gray and CI are in red. Results of a spearman's ranked correlation test are displayed in each graph with significant correlations after a Bonferroni correction marked with an asterisk and linear regression lines ($p \leq 0.004$).

### GDTs

**GDT_within.** NH listeners performed at floor level on the within-frequency gap detection task with all participants displaying a GDT_within of 2 ms, the shortest gap included in the task. It is possible that the true GDT_within is less than 2 ms in some participants. Previous studies have shown that the GDT_within in NH listeners is affected by stimulus factors such as stimulus frequency, spectral complexity, and marker durations [18, 37, 46]. For instance, using 1 and 2 kHz pure-tone with the marker duration of 10–20 ms, Heinrich and Schneider [19] reported a GDT_within that ranged from 0.9 to 1.6 ms for younger and 1.2 to 2.5 ms for older participants. Using a marker duration of approximately 20 ms, Blankenship et al. [18] reported that 2 kHz pure-tone GDT_within ranged from 2 to 15 ms (Mean = 5.8 ms) for young NH listeners. In contrast, using narrow-band noise of 400 ms in duration at a center frequency at 2 kHz and 1 kHz, Lister et al. [37] reported that GDT_within ranged from 7 to 15 ms (Mean = 9.8 ms) in young NH adults. Using a 2 kHz narrowband noise of approximately 300 ms as the markers,

Alhaidary and Tanniru [46] reported that the same group of NH listeners displayed a mean $GDT_{within}$ of 3.33 and 8.33 ms using different testing programs that incorporated stimuli with only subtle spectral differences. Previous studies have shown that GDTs increase with stimulus complexity and shorter marker durations which could explain the slightly higher GDTs reported by previous studies [18, 37, 46].

CI users displayed a $GDT_{within}$ of 2 ms in 12 CI ears and a much longer $GDT_{within}$ (Range = 27–52 ms) in 3 CI ears. The $GDT_{within}$ from CI users did not differ statistically from that in NH peers. In our previous study using a shorter marker duration (~20 ms), 2 kHz pure-tone $GDT_{within}$ ranged from 5 to 70 ms in CI users (Mean = 23.2 ms; [18]). Using noise centered at 500 Hz as the markers, Tyler et al. [20] reported CI users displayed a $GDT_{within}$ that ranged from 7.5 to 300 ms. Using narrowband noise and various stimulus durations, Muchnik et al. [21] reported $GDT_{within}$ in CI users that ranged from 12–72 ms, with smaller GDTs as the stimulus duration increased. Using electrical stimulation, Shannon [66] reported a $GDT_{within}$ in CI users that is comparable to the reported $GDT_{within}$ in NH listeners when the stimulus intensity was comparable. The authors suggested that CI users as a group may not necessarily have poorer temporal resolution. Mussoi and Brown [42] reported a median $GDT_{within}$ in a group of CI users that was less than 3 ms. The slightly higher mean GDTs reported in the literature could be due to several factors including the complexity of the signal, overall stimulus duration, and inclusion of pre-lingually deafened CI recipients. Overall, our NH and CI $GDT_{within}$ results are consistent with the literature and further supports the variability seen in CI recipients.

**$GDT_{across}$**. The difference between NH and CI users in $GDT_{across}$ scores (NH = 58.8 ms, CI = 82.4 ms), regardless of the wide variability was not found to be significant. $GDT_{across}$ in NH participants have been reported in several previous studies [8, 13, 22–25, 37, 39]. For example, using 2 kHz to 1 kHz pre- to post-gap narrowband noise markers, Lister et al. [39] reported that the $GDT_{across}$ has a mean value of 29 ms in young adults and 56 ms in older adults. These studies have documented wide variability in $GDT_{across}$ and that the $GDT_{across}$ can be ten times poorer than the $GDT_{within}$ [13, 46]. The difference in $GDT_{across}$ obtained in the current study and compared to the literature could be due to differences across studies regarding participant age, instrumentation, stimulus, and presentation levels.

There are only a few $GDT_{across}$ studies in the literature involving CI users, which were conducted with direct electrical stimulation. Hanekom and Shannon [14] measured $GDT_{across}$ using 200 μs/phase biphasic pulses with a stimulation rate of 1000 pps in three adult CI users as a function of electrode separation between the pre- and post-gap markers. The $GDT_{across}$ systematically increased as the channel separation increased with thresholds varying significantly across participants (Range = 10–200 ms). Upon examining the $GDT_{across}$ using electrode pairs corresponding to the frequencies used in the current study, $GDT_{across}$ was approximately 7, 25, and 50 ms in these 3 participants (Mean = 48.7 ms). Using a similar approach, van Wieringen and Wouters [28] examined the $GDT_{across}$ in four post-lingually deafened CI users. While one participant displayed a low $GDT_{across}$ (<10 ms), the other participants had higher $GDT_{across}$ (10 to 50 ms), with a longer $GDT_{across}$ for a larger electrode separation. In the current study, the mean $GDT_{across}$ in the CI group was 82.4 ms (Range = 25–120 ms), which is higher than most individual $GDT_{across}$ reported in previous studies using electrical stimulation [14, 28]. However, the current study used acoustic which are expected to be poorer than those measured with direct electrical stimulation.

Age effect on the $GDT_{across}$ has been observed in the current study, with a much higher $GDT_{across}$ in older (> 50 years of age) than younger participants for both CI group (Younger = 65 ms, Older = 98 ms) and NH group (Younger = 31 ms, Older = 83 ms). This negative effect of participant's age is consistent with studies by previous studies, with more obvious

effect for the GDT$_{across}$ [12, 19, 37, 39, 67]. In summary, the GDT$_{within}$ and GDT$_{across}$ reported in the literature and in this study showed some differences, which may be attributed to different factors in the stimuli used and participants' age. The stimulus factors include stimulus level, marker bandwidth, marker duration and frequency, difference in the pre- and post-gap markers in spectral composition and intensity, and stimulus presentation mode [11, 12, 15, 16, 31, 37, 68]. Both GDT$_{within}$ and GDT$_{across}$ show a large variability, with some CI users displaying GDTs comparable to those in NH listeners while others had much higher GDTs [4, 28].

## CAEPs

**CAEPs to within-frequency gap stimuli.** NH listeners displayed a trend of increased N1-P2 amplitude for longer gap durations (sub-threshold = 1.2 µV, threshold = 1.5 µV, supra-threshold = 1.8 µV, see Fig 3 and Table 2). This finding is similar to that reported in previous studies [37, 39]. Compared to NH listeners, CI users' CAEPs showed the following differences: 1) both pre-gap and post-gap markers displayed poorer morphology and smaller N1-P2 amplitude relative to NH listeners. This might be due to a reduced number of neurons that are able to respond synchronously to the markers as a result of deafness effects in CI users. 2) CI users did not display a trend observed in NH listeners that N1-P2 amplitude increases with the increase in the gap duration. This may be explained with the neural recovery mechanism involved in gap detection [37–39]. Specifically, for within-frequency gap stimuli, the neurons initially respond to the pre-gap marker and then are stimulated again by the post-gap marker. In NH listeners, the neural activities are more likely to recover from the pre-gap stimulation if the gap duration is longer. However, CI users have compromised neural firing and synchronization and their neurons may have a large variability in the recovery time. Therefore, their post-gap CAEP is typically poor and does not show the progressive change with the change of the gap duration.

**CAEPs to across-frequency gap stimuli.** Compared to the CAEP to the within-frequency gap stimuli, the CAEP evoked by the across-frequency gap stimuli show the following differences: 1) all post-gap CAEPs have similar amplitude for all gap durations in both NH and CI groups (see Fig 4 and Table 3). 2) the post-gap CAEP appears to be slightly larger than the pre-gap CAEP in both NH and CI groups. 3) the NH vs. CI difference in the post-gap CAEP is less dramatic in its amplitude for the across-frequency gap stimuli than for the within-frequency gap stimuli (see Fig 3 vs. Fig 4). This is because the across-frequency gap stimuli elicit responses of different groups of neurons as the pre- and post-gap frequencies are different and thus the post-gap CAEP is similarly synchronized compared to the pre-gap CAEP.

A limited number of studies have examined the CAEP to across-frequency gap stimuli, all of which were conducted with individuals with normal or minimal hearing loss [37, 39]. Lister and colleagues [37, 39] examined the CAEP to across-frequency gap using narrowband noise stimuli (2 kHz pre-gap and 1 kHz post-gap marker) in a group of young and older adults. The authors included different gap durations conditions including threshold, sub-threshold, and supra-threshold conditions, and a reference condition (a standard 1 ms gap). Results showed the CAEP amplitude was similar for all gap durations. Similar to the findings from Lister and colleagues, the current study did not find significant effects of gap duration condition on CAEP amplitude to the across-frequency gap stimuli. Age effect on the CAEP has been observed in this study. Individuals younger than 50 years of age displayed shorter N1 and P2 latencies (N1 = 95.3 ms, P2 = 151.1 ms) compared to older individuals (N1 = 99.2 ms, P2 = 171.4 ms). Similarly, Lister and colleagues [37, 39] reported an age effect on the P2 latency.

## Different mechanisms for within- and across-frequency gap detection

Based on the CAEP and GDT findings in this study, we propose that different mechanisms may be involved in detecting within- and across-frequency gaps. For the within-frequency gap detection, the listeners rely on the comparison of sensory encoding of the pre- and post-gap markers of the same frequency. When the gap is long enough, the post-gap neural activity will be strong enough [68], the sensation for the post-gap marker will be recovered [15], and the participant can detect the gap. For the across-frequency gap detection, the post-marker CAEP is nearly similarly strong for all gap conditions because the post-marker CAEP is mainly evoked by the acoustic changes consisting of the frequency change (from 2 kHz to 1 kHz in this study) and the gap. Therefore, this post-gap CAEP is a type of the acoustic change complex (ACC) that increases the amplitude with the increase of salience of the sound change in gap and frequency [42, 59, 63, 69]. Although the participants were instructed to just pay attention to the silent gap and ignore the frequency difference between the pre- and post-gap markers, the latter may have acted as a distractor so that the participants cannot fully engage their cognitive attention to perform the across-frequency gap detection and produced a much larger value for $GDT_{across}$ than $GDT_{within}$. Therefore, the across-frequency gap detection involves more central mechanism than the within-frequency gap detection. The involvement of cognitive function and allocation of attentional resources in gap detection has been suggested in previous behavioral studies [13, 28]. With the CAEP and behavior measures, this study provided more evidence that there may be different neural mechanisms underlying within- and across-frequency gap detection, with the later involving more cognitive mechanisms.

## Correlation analysis

**$GDT_{within}$ vs. speech perception.** Previous studies have shown a correlation between the $GDT_{within}$ and speech perception in CI users, with a much shorter stimulus duration compared to that used in this study [18, 21, 70]. Our data using pure-tone markers of approximately 300 ms showed that 3 CI ears with the longest GDTs (Sci36-Left = 51.7 ms, Sci36-Right = 26.7 ms, Sci43-Left = 41.7 ms) had a much worse AzBio-Noise performance compared to other CI users. Examining demographic and device data in Table 1, Sci36 is the oldest participant in the study with the longest duration of auditory deprivation (51 yrs.). Sci43 is also an older participant (60 yrs.) with 23 years of auditory deprivation. With regard to speech perception, Sci43 displayed the poorest performance in the CNC-Word (43%) and Phoneme (62%) and Sci36 displayed relatively high performance on the CNC-Word (Left = 80%, Right = 74%) and Phoneme (Left = 93%, Right = 86%). On the AzBio sentences in quiet, their scores ranged from 82% to 89%. However, on the AzBio sentences in noise and the BKB-SIN, these two participants were among the poorest performers with AzBio-Noise scores ranging from 40 to 56% and SNR-50 scores of +10.8 to +12.8 dB. Our results indicate that adequate speech understanding in quiet may be achieved with $GDT_{within}$ greater than 25 ms, but smaller $GDT_{within}$ (i.e., better temporal processing) is needed to perform speech-in-noise tasks.

The lack of variability in the $GDT_{within}$ and behavioral speech perception partially undermined correlation analysis between these two measures. A shorter marker duration in gap detection is more likely to separate individuals with normal and abnormal temporal resolution [68]. Using tone stimuli of durations of approximately 20 ms in CI users, Blankenship et al. [18] reported GDTwithin ranged from 2 to 100 ms and the GDTwithin were significantly correlated with speech perception performance in quiet and background noise. Tyler et al. [20] reported that CI recipients with GDTs > 40 ms displayed poorer speech and environmental noise identification abilities compared to CI recipients with GDTs < 40 ms. Similarly, Muchnik et al. [21] reported mean narrowband noise GDTs of 12.2 ms (SD = 18.7) in CI recipients

with open-set speech recognition and a mean of 41.0 ms (SD = 34.3) in participants without significant open-set speech recognition.

**$GDT_{across}$ vs. speech perception.**   Temporal cues in natural speech such as the voice onset time and silent gaps occur between sounds of various spectral compositions. Therefore, it is reasonable to assume that the $GDT_{across}$ is more related to behavioral speech perception than the $GDT_{within}$. Previous studies examining the correlation between the $GDT_{across}$ and speech perception performance have reported mixed results. Elangovan and Stuart [71] showed a significant correlation (ranging from approximately 0.55 to 0.75) between voice onset time phonetic boundaries and the $GDT_{across}$ rather than the $GDT_{within}$. In contrast, Mori et al. [72] did not find a significant correlation between voice onset time boundaries or slope (/ba/ to /da/) and the $GDT_{across}$ in a group of Japanese listeners. In the current study, our results did not reveal a correlation between clinical measures of speech perception and the $GDT_{across}$ ($p$ >0.01). Our results suggest that the $GDT_{across}$ is not a good indicator of clinical measures of speech perception.

**CAEP vs. speech perception.**   This study found a significant correlation between within-frequency CAEP measures and speech perception in quiet (CNC words) and noise (AzBio and BKB-SIN sentences). Individuals with poorer speech performance had a smaller N1-P2 amplitude and longer N1 latency. No correlations were found between across-frequency CAEP measures and speech perception performance. Overall correlation analyses indicate that the CAEP evoked by the within-frequency gap stimuli is a promising objective tool that can be used to predict CI speech outcomes.

Only a few other studies have explored how the CAEP to gaps may relate to speech perception using bivariate correlations or descriptive summaries [30, 32, 41]. Michalewski et al. [32] reported that individuals with auditory neuropathy (9–60 yrs) had elevated behavioral GDTs and poorer speech understanding (sentences in quiet), compared with normal hearing young adults (18–30 yrs). Additionally, CAEPs were presented for conditions with longer gap durations. The GDT measured with behavioral and electrophysiological methods were similar in both auditory neuropathy group and normal hearing group. The researchers did not examine the correlation between CAEP measures and speech perception performance. In pediatric CI recipients with auditory neuropathy, He et al. [41] reported a significant negative correlation between electrophysiological GDTs (800 ms biphasic electric pulses with silent gaps) and PBK word scores. Furthermore, in CI candidates with auditory neuropathy, He et al. [30] reported a significant negative correlation between electrophysiological GDTs and word scores. However, it is important to note that the CAEP values used in the correlation analyses were electrophysiological GDT (ms) and not P1-N1-P1 peak amplitude and latency values.

Most studies examining the correlation between CAEP and speech perception have focused on the CAEP evoked by stimulus onset (onset-CAEP) rather than the CAEP evoked by the gap (a change in a longer stimulus). These studies have shown a relationship between speech perception performance and P1-N1-P2 response amplitude and latency. Specifically individuals with individuals with poorer speech identification scores showed a significantly smaller CAEP amplitude [73, 74] increased N1 latency [75, 76] and poorer CAEP morphology [74, 77] compared to CAEP responses from individuals with good speech perception performance. Differences in the CAEP responses between individuals with good vs. poor speech perception performance have been proposed to be due to a combination of factors surrounding neural integrity (density, synchronization, adaptation, and refractory periods) [36, 78–80]. For example, smaller CAEP amplitude and longer latencies may be due to a reduced number of neurons that are able to respond to acoustic stimuli due to auditory deprivation. This explanation can also be used to explain the current finding that individuals with poorer speech perception performance had poorer within-frequency gap-evoked CAEPs. Additionally, the cortical neurons

have an increased refractory period due to long-term deafness and are not able to fire as quickly following the pre-marker stimuli.

## Implications and limitations

Our results indicate within-frequency gap detection, which is related to speech perception performance, is a better approach than the across-frequency gap detection to examine temporal resolution. CAEP to within-frequency gap stimuli could be used an objective measure of temporal resolution. However, there are several limitations of this study. The first limitation is that there was a floor effect observed on the $GDT_{within}$ in both NH listeners and CI users, the smallest possible value on the task. Future behavioral gap detection studies should shorter marker durations, thereby increasing the difficulty level of the task and avoiding the floor effects of the $GDT_{within}$. Similarly, there was a ceiling effect observed for NH listeners for several of the speech perception tasks. Future studies should be designed so that the task difficulty between the NH and CI listeners is equivalent. Additionally, the rise/fall time around the silent gap was 1 ms, as in previous studies [37], which may cause acoustic transients and energy spread [81] that affect the GDT and CAEP results. However, such a spectral spread should be the same for both within- and across-frequency gap stimuli used in this study. Therefore, the different trends of the post-gap CAEPs for the within- and across-frequency stimuli still enabled us to infer different neural mechanisms involved in the within- and across-frequency gap detection. Future studies will use approaches suggested in previous studies, e.g., by using notched noise to mask the transient cue, for gap detection tasks [7, 10].

## Conclusions

CI users had significantly poorer speech performance compared to age-matched NH listeners but corresponding group difference in behavioral GDTs was not observed. The 3 CI ears (two CI users) with the worst $GDT_{within}$ performed poorer in AzBio-Noise than other CI users whose $GDT_{within}$ was comparable to that in NH listeners. The CAEP to within-frequency gap stimuli in NH listeners showed the anticipated trend of increased N1-P2 amplitude with the increase of the gap duration. This trend was weak in CI users, likely due to compromised neural firing and increased neural recovery time. The CAEP to across-frequency gap stimuli displayed a similar amplitude for all tested gap durations in both NH and CI groups. This is because the post-gap stimuli elicits a different group of neurons compared to the pre-gap stimuli. Older individuals displayed larger GDTs and had poorer CAEP responses (smaller amplitude and delayed latencies). Additionally, within-frequency CAEP responses were significantly correlated with open-set word and sentence recognition. Therefore, our findings support that $GDT_{within}$ rather than the $GDT_{across}$ is a better indicator for temporal resolution and the CAEP to within-frequency gap stimuli can serve as an objective tool to predict speech perception performance.

## Supporting information

**S1 Table. NH and CI group mean GDTs and speech perception performance.**
(DOCX)

**S2 Table. Behavioral mixed effect model analysis (*p*-values, F-test, degrees of freedom displayed).**
(DOCX)

**S3 Table. Within and across-frequency post-gap CAEP mixed effect model analysis (*p*-values, F-test, degrees of freedom displayed).**
(DOCX)

## Acknowledgments

Portions of this article were presented at the Association for Research in Otolaryngology Annual MidWinter Meeting (2014, 2019). This manuscript represents a portion of the doctoral dissertation of the first author under guidance of the co-authors. Additionally, this manuscript is available on MedRxiv as two separate preprints, one focused on within-frequency and the other on across-frequency results. The content is solely the responsibility of the authors and does not necessarily represent the official views of the National Institutes of Health or the University of Cincinnati.

## Author Contributions

**Conceptualization:** Chelsea M. Blankenship, Fawen Zhang.

**Data curation:** Chelsea M. Blankenship.

**Formal analysis:** Chelsea M. Blankenship, Jareen Meinzen-Derr.

**Funding acquisition:** Chelsea M. Blankenship, Fawen Zhang.

**Investigation:** Chelsea M. Blankenship, Fawen Zhang.

**Methodology:** Chelsea M. Blankenship, Fawen Zhang.

**Project administration:** Chelsea M. Blankenship.

**Resources:** Fawen Zhang.

**Supervision:** Fawen Zhang.

**Validation:** Chelsea M. Blankenship.

**Visualization:** Chelsea M. Blankenship.

**Writing – original draft:** Chelsea M. Blankenship.

**Writing – review & editing:** Chelsea M. Blankenship, Fawen Zhang.

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
