## [Decision Letter · Decision Letter 0]

17 Jun 2022

PONE-D-22-13719Within- and Across-Frequency Temporal Processing and Speech Perception in Cochlear Implant UsersPLOS ONE

Dear Dr. Blankenship,

Thank you for submitting your manuscript to PLOS ONE. After careful consideration, we feel that it has merit but does not fully meet PLOS ONE’s publication criteria as it currently stands. Therefore, we invite you to submit a revised version of the manuscript that addresses the points raised during the review process.

We look forward to receiving your revised manuscript.

Kind regards,

Prashanth Prabhu

Academic Editor

PLOS ONE

Journal Requirements:

"This research was supported by the NIH NIDCD R15 DC011004 (FZ) and the University of Cincinnati Research Council (CMB)."

Reviewers' comments:

Reviewer's Responses to Questions

**Comments to the Author**

1. Is the manuscript technically sound, and do the data support the conclusions?

Reviewer #1: Partly

Reviewer #2: Yes

2. Has the statistical analysis been performed appropriately and rigorously? 

Reviewer #1: Yes

Reviewer #2: I Don't Know

3. Have the authors made all data underlying the findings in their manuscript fully available?

Reviewer #1: Yes

Reviewer #2: Yes

4. Is the manuscript presented in an intelligible fashion and written in standard English?

Reviewer #1: Yes

Reviewer #2: Yes

5. Review Comments to the Author

Reviewer #1: The authors made an attempt to study the behavioral and electrophysiological GDT in normal hearing and CI users and compared it speech perception in quiet and noise. The design of the study is appreciable. However, following queries needs to be addressed before considering for publication.

1. Line 116: Ambiguous sentence

2. Line 193: The reason for taking 2kHz and 1 kHz in particular was not justified.

3. Line 223: The authors used different transducers for NH and CI for behavioural GDT which was not justified.

4. Line 240-241: The authors need to clarify how was the reference (no gap) stimuli constructed for GDT across for 2AFC.

5. The soundcard details of the personal computer is not provided. Further, were the stimuli routed through audiometer? If so, was the details of the audiometer to be given.

6. Pictorial representation of the stimuli for GDT needs to be provided for better understanding.

7. Test-retest reliability is not checked for this study. Authors need to clarify.

8. The mean thresholds of CI users at 4 kHz and 6 kHz are higher. This is unusual as most of the time the thresholds are around 25-30 dB even for high frequencies in the CI users.

9. The discussion for CAEP vs Speech performance is inadequate.

Reviewer #2: Overall the methods of the study were clear and the objectives were accomplished. The literature review was extensive and covered the topic thoroughly. The individual discussion sections on the individual effects seem to be orphaned from the final concluding remarks and conjecture (the traditional discussion section) and read more as an extension of the results section with a comparison to literature more than providing rationale in literature and then explaining the significance of the correlation. A reorganization of text is suggested for the discussion section, however, all of the data and conjecture is there. No additional information is needed. There is a great deal of supplementary material that may be better served as 1-2 reference lines in the results section to include the mean/std of the CEAP data. Lastly, there were a lot of various statistical analyses performed. While not my expertise, a direct comparison of individual effects using the mixed effects model could be better explained. The study was well written and clear and of significance to the field of cochlear implants and speech science.

Minor Comments:

Line 94: Do you mean, “markers”?

Line 102: Typically, “stimulation mode” refers to the nature of the grounding of the electrical circuit or the type of the pulse (monopolar, tripolar, etc.). Perhaps use “experimental modalities” or “presentation modalities”

Line 104: Suggest using naturalistic audio instead of “daily life sounds”

Lines 103-105: Does this sentence mean to say that the resulting response of the stimuli provides a better response with the clinical processor and microphone than directly stimulating the intracochlear array? Is there any literature, or have the authors performed this task, regarding the comparison of the same task across different presentation approaches? Ideally these experiments take place inside a sound booth, in a controlled manner which would leave the researcher to assume the input from the microphones and the input sent to the clinical processor directly would be the same.

Line 112: Check the in-text reference to [20] to ensure it matches the other references

Line 116, Lines 428-429: Preferential use of first person pronoun, suggest stating “the authors of the present study reported that…”

Line 121: “increases”

Line 145: Check the in-text reference style

Line 185: “marker”

Line 212: Check the in-text reference style

Line273: “Behavioral data” can mean a wide variety of thing, typically used for psychology-based experiments. Perhaps use “subjective data” or “performance” – the authors use “speech perception” which would also be appropriate

Line 302: “makers”

Line 332: Remove the “a” before “much higher”

Figure 2: Typically this is useful for validation purposes such as dissertations, but may not be needed here and could save space in the journal (to potentially include some of the supplemental data here instead)

Line 396: “Bonferroni corrections were applied…”

Line 397 and Line 404: “…therefore a p-value of 0.01 was…”

Line 403-404: Are the authors using the post-hoc, multiple comparisons tests using Bonferroni corrections? These explicit comparison groups using MC analysis should be more clear in the methods section

Lines 423-424: The difficult of the task between CI and NH users should be equivalent such that there are no ceiling level effects for either group, especially for NH users. In literature, many CI studies which use NH data will lower the SNR for the NH group while keeping the CI SNR somewhat higher. While the tests cannot be completed again, a stronger correlation could be had when this factor is controlled between groups.

Line 425-426: Did the authors consider the performance level of the CI group? For example, splitting the data into groups based on their performance level in quiet with their clinical processor. This type of analysis could also provide insight on the large variability seen with CI studies. Although the study was age-matched, it could be more impactful if the study was also matched based on performance. For CI users, this is incredibly difficult to do, but would be a worth-while comparison at the end of the day.

Line 429: A period is needed after the (18, 46,62) reference

Line 429, Line 452-453: “In a previous study by the authors,… a mean CNC-Word score of … was reported…”

Line 452: Include the take-away from the previous studies on GDTwithin or remove the sentence. This is covered in the introduction (or if new literature, should be added)

Line 454: Remove “a” before noise, or include stimuli after “a”

Lines 450-464: The correlation of the GDTwithin and speech perception of the present study is not clear here or how it relates to the mentioned literature. Please elaborate. A better comparison is needed.

Line 452: “shorter”

Line 465: “The difference between NH and CI users in GDTacross scores, regardless of the wide variability was not found to be significant.”

Lines 489-495: include with previous paragraph

Line 536: “Markers”

6. PLOS authors have the option to publish the peer review history of their article (what does this mean?). If published, this will include your full peer review and any attached files.

Reviewer #1: No

Reviewer #2: No

---

## [Author Response · Author response to Decision Letter 0]

17 Aug 2022

See Response to Reviewers document that was uploaded.

---

## [Decision Letter · Decision Letter 1]

13 Sep 2022

PONE-D-22-13719R1Within- and across-frequency temporal processing and speech perception in cochlear implant usersPLOS ONE

Dear Dr. Blankenship,

Thank you for submitting your manuscript to PLOS ONE. After careful consideration, we feel that it has merit but does not fully meet PLOS ONE’s publication criteria as it currently stands. Therefore, we invite you to submit a revised version of the manuscript that addresses the points raised during the review process.

 Please submit your revised manuscript by Oct 28 2022 11:59PM. If you will need more time than this to complete your revisions, please reply to this message or contact the journal office at plosone@plos.org. Please include the following items when submitting your revised manuscript:A rebuttal letter that responds to each point raised by the academic editor and reviewer(s). You should upload this letter as a separate file labeled 'Response to Reviewers'.A marked-up copy of your manuscript that highlights changes made to the original version. You should upload this as a separate file labeled 'Revised Manuscript with Track Changes'.An unmarked version of your revised paper without tracked changes. You should upload this as a separate file labeled 'Manuscript'.If applicable, we recommend that you deposit your laboratory protocols in protocols.io to enhance the reproducibility of your results. Protocols.io assigns your protocol its own identifier (DOI) so that it can be cited independently in the future. For instructions see: https://journals.plos.org/plosone/s/submission-guidelines#loc-laboratory-protocols. Additionally, PLOS ONE offers an option for publishing peer-reviewed Lab Protocol articles, which describe protocols hosted on protocols.io. Read more information on sharing protocols at https://plos.org/protocols?utm_medium=editorial-email&utm_source=authorletters&utm_campaign=protocols.

We look forward to receiving your revised manuscript.

Kind regards,

Prashanth Prabhu

Academic Editor

PLOS ONE

Journal Requirements:

Additional Editor Comments:The authors have incorporated most of the suggestions provided by the reviewers. There are few minor concerns still which should be addressed before its considered for publication. 

Reviewers' comments:

Reviewer's Responses to Questions

**Comments to the Author**

1. If the authors have adequately addressed your comments raised in a previous round of review and you feel that this manuscript is now acceptable for publication, you may indicate that here to bypass the “Comments to the Author” section, enter your conflict of interest statement in the “Confidential to Editor” section, and submit your "Accept" recommendation.

Reviewer #1: All comments have been addressed

Reviewer #3: (No Response)

2. Is the manuscript technically sound, and do the data support the conclusions?

Reviewer #1: Yes

Reviewer #3: Yes

3. Has the statistical analysis been performed appropriately and rigorously? 

Reviewer #1: Yes

Reviewer #3: Yes

4. Have the authors made all data underlying the findings in their manuscript fully available?

Reviewer #1: Yes

Reviewer #3: Yes

5. Is the manuscript presented in an intelligible fashion and written in standard English?

Reviewer #1: Yes

Reviewer #3: Yes

6. Review Comments to the Author

Reviewer #1: The discussion for CAEP and speech perception needs to be revised further. The discussion should explain the presence of smaller N1-P2 amplitude and longer N1 latency and its significance to poor speech perception.

Reviewer #3: General Comments

• Well thought research experiment.

• As rightly mentioned in the limitations, considerations of the smaller gaps and rise time of the GDT stimulus would have further strengthen the methods.

• Explanations provided by the authors are fine but already existing and not conclusive (It cannot be either with the study undertaken). However, the need is to define the actual processes responsible for these phenomena.

Introduction

1. Line 148 to 156: Relation of CAEP peak’s amplitude and latency with the change in GDT – stimuli – paradigm should be mentioned more clearly in the Introduction (in NH population, especially) to provide a better idea in the relationship to readers naïve to this topic.

Methods

2. Most of the stimulus presentation is via the speaker. Information on the make of the speaker to be added.

3. Line 242 to 245: What was the need of occluding the contralateral ear while estimating Speech Perception in CI patients? Just switching off the processor of that side would have done it.

4. GDT involves the processing of the shortest gap which is more dependent on cortical processing. Therefore, to me, having 2 ears of the same person in the study is not right. Because you will be tapping the same cortex. Justify, please.

Results

5. Sample of the study is very small, therefore the inclusion of the analysis of the effect of age and ear is inappropriate.

Discussion

6. Line 783: Insert comma after Additionally.

7. PLOS authors have the option to publish the peer review history of their article (what does this mean?). If published, this will include your full peer review and any attached files.

Reviewer #1: No

Reviewer #3: **Yes: **UDIT SAXENA

---

## [Author Response · Author response to Decision Letter 1]

22 Sep 2022

Journal Requirements:

I have reviewed the reference list to ensure that it is complete and correct.

Additional Editor Comments: 

The authors have incorporated most of the suggestions provided by the reviewers. There are few minor concerns still which should be addressed before its considered for publication. 

Reviewers' comments:

Reviewer's Responses to Questions 

Comments to the Author

1. If the authors have adequately addressed your comments raised in a previous round of review and you feel that this manuscript is now acceptable for publication, you may indicate that here to bypass the “Comments to the Author” section, enter your conflict of interest statement in the “Confidential to Editor” section, and submit your "Accept" recommendation.

Reviewer #1: All comments have been addressed

Reviewer #3: (No Response)

2. Is the manuscript technically sound, and do the data support the conclusions?

Reviewer #1: Yes

Reviewer #3: Yes

3. Has the statistical analysis been performed appropriately and rigorously? 

Reviewer #1: Yes

Reviewer #3: Yes

4. Have the authors made all data underlying the findings in their manuscript fully available?

Reviewer #1: Yes

Reviewer #3: Yes

5. Is the manuscript presented in an intelligible fashion and written in standard English?

Reviewer #1: Yes

Reviewer #3: Yes

6. Review Comments to the Author

Reviewer #1: The discussion for CAEP and speech perception needs to be revised further. The discussion should explain the presence of smaller N1-P2 amplitude and longer N1 latency and its significance to poor speech perception.

We have added information about the relationship between CAEP and speech perception performances to lines 671-700. 

Reviewer #3: General Comments

• Well thought research experiment.

• As rightly mentioned in the limitations, considerations of the smaller gaps and rise time of the GDT stimulus would have further strengthen the methods.

• Explanations provided by the authors are fine but already existing and not conclusive (It cannot be either with the study undertaken). However, the need is to define the actual processes responsible for these phenomena. 

Introduction

1. Line 148 to 156: Relation of CAEP peak’s amplitude and latency with the change in GDT – stimuli – paradigm should be mentioned more clearly in the Introduction (in NH population, especially) to provide a better idea in the relationship to readers naïve to this topic.

We have added more information about the effect of gap duration on CAEP amplitude and latency to lines 140-169.

Methods

2. Most of the stimulus presentation is via the speaker. Information on the make of the speaker to be added.

We have added information about the speakers to lines 247 and 261.

3. Line 242 to 245: What was the need of occluding the contralateral ear while estimating Speech Perception in CI patients? Just switching off the processor of that side would have done it.

Yes for bilaterally implanted CI recipients we removed the contralateral speech processor. However some CI recipients were unilaterally implanted. Therefore to keep it consistent across all participants we removed contralateral devices if applicable and then occluded the contralateral ear during testing. 

This information is already contained on Lines 235-238: “For NH individuals and unilateral CI users, testing was completed with the contralateral ear occluded with an E-A-R disposable foam ear plug. For bilateral CI users, testing was completed with the contralateral speech processor removed.”

4. GDT involves the processing of the shortest gap which is more dependent on cortical processing. Therefore, to me, having 2 ears of the same person in the study is not right. Because you will be tapping the same cortex. Justify, please.

I agree with the reviewer that the responses from the left and right ear of the same participant are not going to be completely independent of each other. This is why we included participant ID as a random effect in the model to account for similar variabilities in ears “clustered” around an individual. Furthermore, while I agree that cortical processing is important, there are other sub-cortical nuclei within the central auditory nervous system are critical for temporal processing. For example, the inferior colliculus contain neurons that are sensitive to stimulus duration, essential to the perceptual ability to detect temporal changes in the stimulus, such as gap detection or amplitude modulation [1]. Animal studies of single neurons within the mouse inferior colliculus have demonstrated electrophysiological GDTs that are within 1 to 2 ms of behavioral GDTs [2]. So there are ear specific aspects of temporal processing that are critical to gap detection thresholds and CAEPs. 

This is also apparent in the raw data. There are some participants with behavioral GDTs that have very similar thresholds for the left and right ear while others show differences in thresholds up to > 20 ms. The same holds true for the CAEP amplitude and latency values. For example, some participants have left and right N1-P2 amplitude values that are very similar (0.98 vs 1.03 µV), others display large differences in amplitude (1.29 vs 3.64 µV). So the results across ears are not necessarily identical and we have taken steps to factor the correlation in responses into our analysis. 

Results

5. Sample of the study is very small, therefore the inclusion of the analysis of the effect of age and ear is inappropriate.

We admit that the small sample size is a limitation of this study. Thus, we worked closely with an experienced biostatistician (Jareen Meinzen-Derr) for both study planning and statistical analyses. Models that were constructed testing the feasibility of the inclusion of age and ear to avoid overfitting the model. Additionally, we are aware of debates regarding sample size in neuroscience studies. Our study included 15 CI ears and 15 NH control ears among 22 participants; such a sample size is not uncommon for EEG studies involving CI users [3-5]. We strongly believe the significant findings of our work will lead to future research efforts of examining temporal processing in CI users with larger sample sizes. 

Discussion

6. Line 783: Insert comma after Additionally.

Change made as suggested. 

7. PLOS authors have the option to publish the peer review history of their article (what does this mean?). If published, this will include your full peer review and any attached files.

Do you want your identity to be public for this peer review? For information about this choice, including consent withdrawal, please see our Privacy Policy.

Reviewer #1: No

Reviewer #3: Yes: UDIT SAXENA

References:

1. Faure PA, Fremouw T, Casseday JH, Covey E. Temporal masking reveals properties of sound-evoked inhibition in duration-tuned neurons of the inferior colliculus. J Neurosci. 2003;23(7):3052-65. Epub 2003/04/10. PubMed PMID: 12684492.

2. Walton JP, Frisina RD, Ison JR, O'Neill WE. Neural correlates of behavioral gap detection in the inferior colliculus of the young CBA mouse. Journal of comparative physiology A, Sensory, neural, and behavioral physiology. 1997;181(2):161-76. Epub 1997/08/01. PubMed PMID: 9251257.

3. Rahne T, Plontke SK, Wagner L. Mismatch negativity (MMN) objectively reflects timbre discrimination thresholds in normal-hearing listeners and cochlear implant users. Brain Res. 2014;1586:143-51. Epub 2014/08/26. doi: 10.1016/j.brainres.2014.08.045. PubMed PMID: 25152464.

4. Wagner L, Plontke SK, Rahne T. Perception of Iterated Rippled Noise Periodicity in Cochlear Implant Users. Audiol Neurootol. 2017;22(2):104-15. Epub 2017/08/30. doi: 10.1159/000478649. PubMed PMID: 28848077.

5. Timm L, Agrawal D, F CV, Sandmann P, Debener S, Buchner A, et al. Temporal feature perception in cochlear implant users. PLoS One. 2012;7(9):e45375. Epub 2012/10/03. doi: 10.1371/journal.pone.0045375. PubMed PMID: 23028971; PubMed Central PMCID: PMCPMC3448664.

---

## [Editor Report · Decision Letter 2]

26 Sep 2022

Within- and across-frequency temporal processing and speech perception in cochlear implant users

PONE-D-22-13719R2

Dear Dr. Blankenship,

We’re pleased to inform you that your manuscript has been judged scientifically suitable for publication and will be formally accepted for publication once it meets all outstanding technical requirements.

Kind regards,

Prashanth Prabhu

Academic Editor

PLOS ONE
---

## [Editor Report · Acceptance letter]

3 Oct 2022

PONE-D-22-13719R2 

Within- and across-frequency temporal processing and speech perception in cochlear implant users 

Dear Dr. Blankenship:

I'm pleased to inform you that your manuscript has been deemed suitable for publication in PLOS ONE. Congratulations! Your manuscript is now with our production department. 

Kind regards, 

on behalf of

Dr. Prashanth Prabhu 

Academic Editor

PLOS ONE